# Bone Fragility in Chronic Kidney Disease Stage 3 to 5: The Use of Vitamin D Supplementation

**DOI:** 10.3390/metabo12030266

**Published:** 2022-03-20

**Authors:** Pablo Antonio Ureña Torres, Jean Claude Souberbielle, Martine Cohen Solal

**Affiliations:** 1Department of Dialysis AURA Nord Saint Ouen, 12, Rue Anselme, 93400 Saint Ouen, France; 2Department of Renal Physiology, Necker Hospital, University of Paris Descartes, 75015 Paris, France; jcsouber91@gmail.com; 3Bioscar INSERM U1132, Department of Rheumatology, Université de Paris, Hôpital Lariboisière, 75010 Paris, France

**Keywords:** fracture, bone mineral density, calcium, phosphate, calcifediol, 25(OH)D, calcitriol, dialysis

## Abstract

Frequently silent until advanced stages, bone fragility associated with chronic kidney disease-mineral and bone disease (CKD-MBD) is one of the most devastating complications of CKD. Its pathophysiology includes the reduction of active vitamin D metabolites, phosphate accumulation, decreased intestinal calcium absorption, renal alpha klotho production, and elevated fibroblast growth factor 23 (FGF23) levels. Altogether, these factors contribute firstly to secondary hyperparathyroidism, and ultimately, to micro- and macrostructural bone changes, which lead to low bone mineral density and an increased risk of fracture. A vitamin D deficiency is common in CKD patients, and low circulating 25(OH)D levels are invariably associated with high serum parathyroid hormone (PTH) levels as well as with bone mineralization defects, such as osteomalacia in case of severe forms. It is also associated with a variety of non-skeletal diseases, including cardiovascular disease, diabetes mellitus, multiple sclerosis, cancer, and reduced immunological response. Current international guidelines recommend supplementing CKD patients with nutritional vitamin D as in the general population; however, there is no randomized clinical trial (RCT) evaluating the effect of vitamin D (or vitamin D+calcium) supplementation on the risk of fracture in the setting of CKD. It is also unknown what level of circulating 25(OH)D would be sufficient to prevent bone abnormalities and fractures in these patients. The impact of vitamin D supplementation on other surrogate endpoints, including bone mineral density and bone-related circulating biomarkers (PTH, FGF23, bone-specific alkaline phosphatase, sclerostin) has been evaluated in several RTCs; however, the results were not always translated into an improvement in long-term outcomes, such as reduced fracture risk. This review provides a brief and comprehensive update on CKD-related bone fragility and the use of natural vitamin D supplementation in these patients.

## 1. Introduction

Renal function and bone mass usually decline parallelly with age, rendering old people more susceptible to suffering from chronic renal dysfunction and osteoporosis [1,2]. The impaired renal function and 1,25 hydroxylase activity led to calcitriol deficiency, amplified by an often-concomitant deficiency of vitamin D substrate [3]. The decrease in calcitriol results in reduced intestinal calcium absorption and hypocalcemia [4]. This hypocalcemia increases the secretion of parathyroid hormone (PTH), stimulates parathyroid cell size and proliferation, and induces a loss of the negative-feedback loop between extracellular calcium and PTH [5]. This phenomenon occurs because the parathyroid cells lose their sensitivity to calcium due to a decreased expression of the calcium-sensing receptor (CaR) [6]. Moreover, hyperphosphatemia progressively appears because of the renal incapacity to excrete and maintain a neutral phosphate balance [7]. High serum phosphate levels also stimulate the production and excretion of PTH by reducing the number of parathyroid CaR and stabilizing the PTH messenger ribonucleic acid (RNA) to promote its synthesis [8]. Finally, the decrease in renal function stimulates the production of fibroblast growth factor 23 (FGF23) by osteocytes under the influence of hyperphosphatemia, which in turn inhibits the synthesis of calcitriol [9]. These alterations leading to secondary hyperparathyroidism (SHPT) and increasing bone turnover have been included in an entity called chronic kidney disease (CKD)-mineral and bone disorders (MBD), with associated laboratory abnormalities, renal osteodystrophy (ROD), and cardiovascular calcification [10,11]. Of note, the bone fragility observed in CKD, which combines either low or high bone turnover, mineralization defects, and low bone mass, can be considered the sum of ROD with age-related osteoporosis [12].

Consequently, the rate of skeletal fracture is significantly increased in CKD patients compared with the age- and gender-matched subjects in the general population [1,13]. These fractures represent a major clinical complication since they are associated with a high risk of mortality and since their prevention and treatment remain a real challenge today [14,15]. As mentioned before, CKD is frequently associated with insufficient/deficient circulating vitamin D (25(OH)D) levels [3], which has been suggested to contribute to the development of SHPT, ROD, bone fragility, fractures, and a large variety of non-skeletal diseases, including cardiovascular, diabetes mellitus, multiple sclerosis, cancer, and reduced immunological response [16]. However, the natural vitamin D dose regimen and the optimal circulating target levels to prevent these complications have been particularly poorly addressed in vitamin D–deficient populations, independent of the presence or absence of CKD [17]. The principal aim of this review is to provide a brief and comprehensive update on CKD-related bone fragility and the use of natural vitamin D supplementation in these patients.

## 2. Epidemiology of Bone Fractures in CKD

CKD-related bone fragility associated with either osteopenia or osteoporosis is two to three times more frequent in CKD patients than in the general population [18,19,20]. Then, in spite of the progress made in the management of CKD and in dialysis techniques, the risk of bone fractures at any skeletal site increases with the severity of CKD, age, and dialysis vintage [1,21,22,23]. The frequency of skeletal fractures can be 2 to 100 times more frequent in CKD patients when compared with age-, ethnicity-, and gender-matched subjects with a normal renal function [14,19,24]. Noteworthy, most of these fractures occur in CKD patients with only osteopenia and not overt osteoporosis, suggesting that other factors such as numerous comorbidities and medications may play an important role [13,14,25,26].

As in the general population, bone fractures in CKD patients are associated with an increased rate of hospitalization and death, particularly in dialysis patients [13,27,28]. The results reported from a UK trial, which included people aged ≥75 years, showed that compared to subjects with an estimated glomerular filtration rate (eGFR) of ≥60 mL/min/1.73 m^2^, the age- and gender-adjusted hazard ratio for hip-fracture related mortality increased from 6% for people with an eGFR from 45–59 mL/min/1.73 m^2^ to 12% for those with an eGFR < 45 mL/min/1.73 m^2^ [29]. Another cohort study using data from Medicare in the USA from 1996 to 2009 showed that the 30-day mortality rate after a hip fracture in incident dialysis patients older than 65 years declined from 20% in 1996 to 16% in 2009 [30]. In France, a study using the 2010 French national health care database showed that following a hip fracture, dialysis patients have a significantly greater mortality rate than those without dialysis, 12% for men and 8% for women. They also have a longer hospital stay in the intensive care unit [13]. In the *Dialysis Outcomes and Practice Patterns Study* (DOPPS), the annual rate of hip fracture in dialysis patients varied from 1.2 to 4.5% according to country [28], with a fracture risk, compared to the general population, that was 1.5 to 8 times higher and a risk of death that was also 3.7 times higher than in the general population. The highest risk of death was seen during the first month following the hip fracture, and the principal causes of death were cardiovascular complications and infection [28].

## 3. Vitamin D and Its Metabolism in CKD

All vitamin D metabolites are found in plasma that is mostly bound to the vitamin D binding protein (DBP), a polymorphic protein with a half-life of 3 weeks that circulates in a high concentration, reducing to a minimum the circulation of any free forms of vitamin D [31,32]. Natural vitamin D undergoes a first hepatic hydroxylation to become 25-hydroxyvitamin D (25(OH)D) (Figure 1). 

Circulating 25(OH)D is then hydroxylated in the proximal tubular cells of the kidney to form 1,25-dihydroxyvitamin D (1,25(OH)2D3) also called calcitriol, which is the active vitamin D metabolite [33]. The renal hydroxylation is stimulated by PTH, low serum calcium and phosphate concentrations, and inhibited by FGF23 and calcitriol itself. Calcitriol binds to a cytosolic receptor, the VDR. Then a trimeric complex associating calcitriol, the VDR, and the retinoic acid receptor (RXR) binds to vitamin D responsive elements (VDRE) in the promoter region of multiple genes to activate or inhibit their transcription. Vitamin D metabolites can be inactivated in the kidney by the 24-hydroxylase enzyme, whose expression is stimulated by FGF23 and calcitriol. The principal effects of the circulating active vitamin D metabolite are the stimulation of intestinal calcium and phosphate absorption, the regulation of bone metabolism, and the control of PTH secretion. There are many other tissues that are able to express the 1,25 hydroxylase and locally transform 25(OH)D into calcitriol. Although most of these tissues are not involved in mineral and bone metabolism, it is worth noting that this extra-renal production of calcitriol also occurs in bone and in the parathyroid cells. In bone, calcitriol is known to exert autocrine/paracrine modulatory effects on osteoblast activity, such as alkaline phosphatase and osteocalcin mRNA upregulation [34], but also in osteoclasts, in which it seems to modulate their resorptive activity [35]. The more recent data argue for a duality in the role of vitamin D metabolites on bone, with long-term beneficial effects in general, but a catabolic effect (i.e., liberating calcium from bone) that aims to maintain calcium balance irrespectively of any adverse effects on the skeleton in the case of calcium deficiency [36]. Finally, relatively recent data from experimental studies suggests that 24,25(OH)2D3, initially thought to be inactive, might be necessary for optimal bone repair after a fracture [37]. This last point, of course, deserves further studies in a clinical human setting, but it is interesting in the context of the present review because the secretion of 24,25(OH)2D3 is known to decrease when renal function declines. In the parathyroid cells, locally produced calcitriol contributes, in association with circulating calcitriol, to suppressing PTH gene expression and to reducing cell proliferation (for an extensive review, see the following: Bienaimé F et al. [38]). This may, in part, explain how vitamin D supplementation may decrease PTH secretion even in dialysis patients with a very low serum calcitriol concentration. It must also be noted that, while extra-renal locally produced calcitriol does not usually contribute to the circulating concentration, it may become significant in CKD anephric patients treated with natural vitamin D. 

In the setting of CKD, circulating 25(OH)D levels begin to decrease from the earliest stages [39,40] due to numerous factors, including skin hyperpigmentation, reduced skin synthesis of cholecalciferol, dietary restriction, impaired intestinal absorption, increased vitamin D catabolism, and important urinary losses of DBP and vitamin D metabolites in the case of severe proteinuria [41,42,43,44,45,46,47]. Furthermore, the renal expression of megalin, the protein responsible for the tubular uptake of 25(OH)D/DBP complex, is decreased in CKD, which exacerbates the already reduced filtration of 25(OH)D/DBP complex because of the CKD and decreased GFR [48,49]. Finally, both SHPT and high FGF23 lead to the stimulation of 24-hydroxylase and the inactivation of vitamin D metabolites [9]. CKD is also characterized by the status of vitamin D hypo-responsiveness because of a progressive loss of VDR in the parathyroid cells [50]. Moreover, low circulating 1,25(OH)2D3 levels impair the formation of the VDR/RXR complex and the binding of trimeric 1,25(OH)2D3/VDR/RXR to the DNA in the VDRE [51,52].

## 4. Circulating Vitamin D Levels and Fractures in CKD

The optimal circulating 25(OH)D level in CKD that is thought to protect against bone fractures is still a matter of debate. The KDIGO (Kidney Dialysis Improving Global Outcome) guidelines recommend assessing circulating 25(OH)D levels at the dialysis initiation, and then once a year or more frequently in case of vitamin D supplementation or in case of evaluating an increased serum PTH level. They also recommend targeting the same 25(OH)D cutoff value as for the general population [53], which is currently 20 ng/mL (or 50 nmol/L) [54,55]. Using another threshold (30 ng/mL) considered more appropriate by some experts [56], the prevalence of vitamin D insufficiency (15–30 ng/mL) and/or deficiency (<15 ng/mL) ranged from 50 to 98%, with a combined prevalence of 82% in different studies performed in CKD patients [40,57]. 

Low circulating 25(OH)D levels are independently associated with a reduced BMD at the radius and at the calcaneus, as demonstrated in the study of 69 hemodialysis patients, where the prevalence of vitamin D deficiency was 59% [58]. Vitamin D insufficiency is also associated with increased subperiosteal bone resorption and with decreased BMD at the lumbar spine and the wrist in CKD dialysis patients [59,60]. Regarding bone fractures, a study performed with 130 non-dialysis CKD patients demonstrated that the patients with bone fractures had significantly lower serum 25(OH)D levels than those without fractures [61]. Another retrospective study with 104 hemodialysis patients showed that those with a serum 25(OH)D level lower than 15 ng/mL, had a decreased bone formation rate and lower trabecular mineralization surfaces compared to a control group, and independently of PTH and calcitriol values [62]. 

## 5. Vitamin D Supplementation and Fractures in CKD

Ensuring adequate vitamin D status is recommended in all guidelines aimed to optimize bone health. In non-CKD subjects, or in subjects not screened according to their renal function, randomized controlled trials (RCTs) have demonstrated that, compared to placebo, supplementation with vitamin D + calcium reduced by 15% (CI: 2–27%) the risk of skeletal fracture at all sites included, and by 30% (CI: 13–44%) the risk of hip fracture, especially in older and vitamin D deficient patients [63]. The effect on fracture risk of vitamin D alone is more controversial and has raised some debate. A large meta-analysis reported no beneficial effect of vitamin D versus placebo or vitamin D + calcium versus calcium [64]. However, this meta-analysis included studies that recruited a relatively low number of participants or used either very low daily doses (400 UI/day) or very high intermittent (300,000–500,000 IU/year) doses, two actions known to have either no effect (very low doses) or even detrimental effects (very high intermittent doses) on the risk of falls and fractures by mechanisms not yet understood [65]. When the analysis was restricted to RCTs that used daily vitamin D doses of 800–1000 IU (8 RCTs), the conclusion was that vitamin D reduced the risk of fracture by 14% (RR 0.86, 95% CI 0.75–0.98) compared to placebo [66]. Last but not least, vitamin D supplementation is considered as essential in osteoporotic patients treated with anti-resorptive drugs, especially bisphosphonates, as it allows a better maintenance of drug efficiency when serum 25(OH)D concentration is maintained above 30 ng/mL [67]. This point is important in the context of the present review, as anti-resorptive drugs, bisphosphonates, or Denosumab, are now considered for the treatment of bone fragility in CKD patients at stage 3–4 [68].

Although the incidence of fractures is higher in CKD patients than in the general population and increases with patient age and CKD severity, no RCTs evaluating the effect of vitamin D (or vitamin D+calcium) supplementation on the risk of fracture are available to our knowledge in these patients. It is thus interesting to look at surrogate endpoints, recognizing, however, that they do not always translate into an improvement in the long-term outcomes, such as fracture risk in patients with CKD. 

KDIGO recommends bone mineral density (BMD) for the evaluation of bone fragility in CKD only if the results will impact therapeutical decisions [53]. Consequently, data on the effect of vitamin D supplementation on BMD in CKD are scarce (Table 1). A post hoc analysis of the DECALYOS II trial results showed that daily supplementation with 800 IU vitamin D3 with 1200 mg of calcium significantly reduced the rate of BMD loss at the distal radius in elderly (mean age, 85 years) institutionalized women with moderate CKD and severe vitamin D deficiency, independently of their baseline serum intact PTH and bone-specific alkaline phosphate levels [69]. In a prespecified secondary endpoint analysis of a RCT, changes in BMD in response to 4000 IU/day vitamin D3 (*n* = 92) or placebo (*n* = 95) for 11 months in incident kidney transplant recipients with a median age of 52 years, a median eGFR of 46 mL/mn/1.73 m² of body surface, and a mean serum 25(OH)D level of 10 ng/mL were evaluated. Forty-four percent of participants had osteopenia or osteoporosis. At the end of the study, the median 25(OH)D level was increased to 40 ng/mL in the vitamin D3 group, and unchanged in the placebo group. The percent change in lumbar spine (LS) BMD from before kidney transplantation to 12 months post-transplantation was −0.2% (95% CI −1.4 to 0.9) in the cholecalciferol group and −1.9% (95% CI −3.0 to −0.8) in the placebo group, with a significant between-group difference (1.7%; 95% CI 0.1 to 3.3), especially in those with the lowest baseline BMD. Changes in BMD at the distal radius were not different between groups [70]. These results, which can only be considered hypothesis-generating and thus deserving further studies, suggest that vitamin D3 supplementation, possibly with calcium, may slow BMD loss in CKD patients with vitamin D deficiency, low bone mass, and beneficiating of a kidney graft (Table 1).

The effect of vitamin D supplementation on bone turnover markers in CKD patients was evaluated in a recent meta-analysis which only studied RCTs [76]. Only two of the included studies reported the following bone marker results: total alkaline phosphatase in one study (one oral 300,000 IU D3 dose or matching placebo in 40 stage 3–4 CKD patients) [71], and bone alkaline phosphatase, serum CTX, and TRAP5b in the other one (50,000 IU D3/week or placebo for 12 weeks in 34 stage 3–4 CKD patients) [72] (Table 1). They reported no difference between the vitamin D3-treated group and the placebo one. Although having less strength in terms of evidence but with a higher number of included patients, we can mention an open-label RCT (60 patients received 2000 IU D3/day and 60 patients received 40,000 IU D3/month for 6 months) [73] and the secondary analysis of the RCT (4000 IU/day vs. placebo discussed above [70] which, similarly, reported no significant changes in bone turnover markers after vitamin D supplementation. By contrast, in a secondary analysis of another RCT in the non-diabetic CKD stage 3–4 patients who received two 300,000 IU oral D3 doses at baseline and 8 weeks (*n* = 58) of a matching placebo (*n* = 59), serum total and bone specific alkaline phosphatase and serum CTX decreased significantly in the D3 group compared to placebo [75], while serum sclerostin concentration was unchanged [75]. 

Contrary to the topics that are discussed above, circulating levels of 25(OH)D negatively correlated with PTH [3,40], numerous studies illustrated the beneficial effect of native vitamin D supplementation on serum PTH concentration in CKD patients. Even in 2011, a meta-analysis of 22 studies (17 observational and 5 RCTs) reported a significant decrease in serum PTH after native vitamin D (cholecalciferol or ergocalciferol) supplementation [77]. The decrease was observed in non-dialysis CKD patients, in dialysis patients, and in renal transplant recipients. These results were consistent with the recommendation made in most guidelines to control serum PTH levels in CKD patients, but due to the high number of observational studies and the small effectiveness of the RCTs included in the meta-analysis, the level of evidence was considered of low-to-moderate quality, deserving more studies, preferably RCTs. Since then, several RCTs have been performed. A meta-analysis of 14 studies (seven placebo-controlled RCTs, two studies that along with the vitamin D supplemented group included a group with no supplementation, and five studies that included only patients that received active supplementation) on the impact of nutritional vitamin D supplementation, either D2 or D3, on serum PTH levels in non-dialysis CKD has been published recently [78]. Interestingly, when estimated as the difference from baseline to the end of the study within the supplemented groups, a small but significant decrease in serum PTH levels was observed, which was unlikely to have by itself any significant clinical effect. However, when comparing the evolution of serum PTH concentration in both supplemented and non-supplemented groups, the authors found a much more significant difference, explained by the fact that PTH increased in the non-supplemented groups. The authors concluded that their results suggested that nutritional vitamin D supplementation in CKD might be effective in preventing further increases in PTH but less effective in reducing PTH in patients with significantly increased levels. The recent secondary analysis of the Japanese RCT in renal transplant recipients discussed above in the BMD paragraph [70], which was not included in this meta-analysis, reported that after 11 months of supplementation with 4000 IU D3/day, serum PTH levels decreased by 39%, from 77 to 40 pg/mL (*p* < 0.01). 

Besides the traditional supplementation with native/nutritional vitamin D (either cholecalciferol or ergocalciferol) and excluding treatment with active 1-hydroxylated vitamin D compounds, supplementation with calcifediol and extended-release (ER) calcifediol has been tested in CKD patients. We have identified one RCT that evaluated the effect of calcifediol given three times a week for six months to 29 CKD stage 3B–4 patients compared to 30 patients that received a matched placebo [79]. The primary endpoint of this trial was changes in pulse-wave velocity (PWV), while changes in biological parameters, including serum PTH concentration, were secondary objectives. The median serum PTH concentration increased from 154.7 to 169.7 pg/mL in the placebo group and decreased from 90.5 to 68.4 pg/mL in the vitamin D group (*p* < 0.001). Several trials tested the effect of ER calcifediol on serum PTH levels. In a first trial published in 2014, the efficacy and safety of a six week course of three different daily doses of ER calcifediol (30, 60, and 90 µg/day) administered to pre-dialysis CKD patients with SHPT and vitamin D insufficiency were compared to placebo [80]. Mean serum PTH levels decreased by 20.9%, 32.8%, and 39.3% in the 30, 60, and 90 μg dose groups, respectively, with no clinically significant safety concerns, and increased by 17.2% in the placebo group (*p* < 0.005). Two identical multicenter RCTs with a total of 429 stage 3–4 CKD patients have been performed after this first publication, the results of which have been pooled in a single article [81]. Patients were randomized 2:1 to receive oral ER calcifediol (30 or 60 μg/day) or placebo for 26 weeks. An extension open-label study was performed for 26 extra weeks in 298 patients, all of whom received ER calcifediol. Serum 25(OH)D levels increased to 53 ng/mL and 68 ng/mL at 12 weeks in those who received 30 and 60 µg/day, respectively, and then remained stable throughout the 52-week treatment period. Patients who achieved a decrease of serum PTH levels of at least 30% were 22, 40, and 50% at 12, 26, and 52 weeks, respectively, and most had a decrease of at least 10%.

As the 25(OH)D serum concentration increases after vitamin D supplementation, reaching sometimes levels that might be considered as alarmingly high such as discussed above about the ER calcifediol trials, safety must be questioned. What is usually feared with too high serum 25(OH)D levels is an increased risk of hypercalcemia, hyperphosphatemia, and kidney stone. The three meta-analyses discussed above [76,77,78] did not report any significant increase in serum phosphate levels with both native vitamin D and ER calcifediol. Kandula and Christodoulou did not report an increase in serum calcium levels, contrary to Bover et al., who reported a slight but significant increase of 0.23 mg/dL after nutritional vitamin D supplementation in non-dialysis CKD patients. This finding was based on seven studies, six that tested vitamin D3 supplementation, and one that tested vitamin D2 supplementation, where serum calcium level was measured at baseline and at the end of the supplementation period [76,77,78] Among these seven studies, three reported an increase of more than 0.2 mg/dL in serum calcium levels. It is worth noting that baseline serum calcium levels were in the low normal range, around 8.8–9.0 mg/dL, in these three studies, suggesting that a significant number of patients were initially hypocalcemic and that they reached the middle of the normal range at the end of the studies, in favor of a normalization of the calcemic status rather than an adverse consequence of vitamin D supplementation. To our knowledge, there are no data indicating an increased risk of kidney stone in the studies that evaluated the effect of vitamin D supplementation in CKD patients (Figure 2).

There are, however, plenty of data on non-CKD patients. In 2006, a 17% increased risk of urolithiasis was evidenced in the Women’s Health Initiative (WHI) study in which 36,282 postmenopausal women aged 50 to 79 years were randomized to receive 1000 mg of elemental calcium with 400 IU of vitamin D3 daily or a matching placebo [82]. Because the vitamin D dose tested in the WHI study was relatively low, this result was questioned by many vitamin D experts who considered that, as the mean baseline dietary calcium intake was already high (1100 mg/day), signifying that the daily calcium intake was thus 2100 mg in the women who received the active treatment, too high calcium intake, rather than vitamin D, was probably the cause of the increased risk of kidney stone. Since then, the results of several “mega-trials” have been very reassuring on the risk of urolithiasis that could be induced by vitamin D supplementation, even in vitamin D-replete subjects. In the VITAL study, 2000 IU vitamin D3 daily for 5.3 years did not induce any increase in the risk of kidney stone in apparently healthy subjects with a very satisfying mean baseline serum 25(OH)D concentration (31 ng/mL) [83]. This was similarly the case observed in the D2D study where prediabetic patients with a baseline serum 25(OH)D concentration of 28 ng/mL received 4000 IU vitamin D3 daily for two years [84], and in the VIDA study, where subjects with a mean baseline 25(OH)D of 24.8 ng/mL received 100,000 IU vitamin D3 monthly for 3.3 years [85]. An exception may concern some patients predisposed to hypercalciuria in whom vitamin D administration might worsen the risk of stone formation [86]. The circulating concentration of FGF-23 increases as renal function declines in CKD patients [87]. Considering the association of FGF23 with adverse outcomes [88], effects of vitamin D3 supplementation on FGF23 concentrations have been evaluated as a secondary endpoint in several RCTs. A recent meta-analysis of eight clinical trials with nine treatment arms did not report any increase in plasma FGF23 levels in patients with CKD after vitamin D supplementation [89]. Taken together, the data discussed above are in favor of a good safety profile of vitamin D supplementation, either native/nutritional vitamin D or ER calcifediol, at the doses that have been tested.

## 6. Conclusions

Insufficient circulating levels of vitamin D are commonly observed in CKD patients, which favors SHPT and fragile bones. Although international guidelines recommend supplementing CKD patients, when circulating 25(OH)D levels are <20 ng/mL, with nutritional vitamin D, as in the general population. Generally, with 800–1400 IU/day of cholecalciferol or ergocalciferol, without surpassing 2000 IU/day, to achieve a serum 25(OH)D levels >20 ng/mL. We believe that this level of circulating 25(OH)D might not be sufficient to prevent PTH raising and bone abnormalities in CKD, as suggested by recent cohort studies where more than two-thirds of CKD patients had controlled serum PTH levels when 25(OH)D levels were higher than 40 ng/mL. Similar findings have also been reported following supplementation with ER calcifediol. Achieving such circulating 25(OH)D levels (50–75 ng/mL) seems to be safe, since they do not increase the risk of hypercalcemia, hyperphosphatemia, and urolithiasis, but in these studies the risk of cardiovascular calcifications was not evaluated. Additionally, as stated by KDIGO guidelines, to reduce the risk of vascular calcification, CKD patients with hyperphosphatemia and hypercalcemia should have calcitriol or another vitamin D sterol reduced or stopped. Therefore, the promising results observed with ER calcifediol need further investigation into the impact of different natural vitamin D compounds on the risk of fractures, CKD progression, cardiovascular calcifications, and mortality. 

## Figures and Tables

**Figure 1 metabolites-12-00266-f001:**
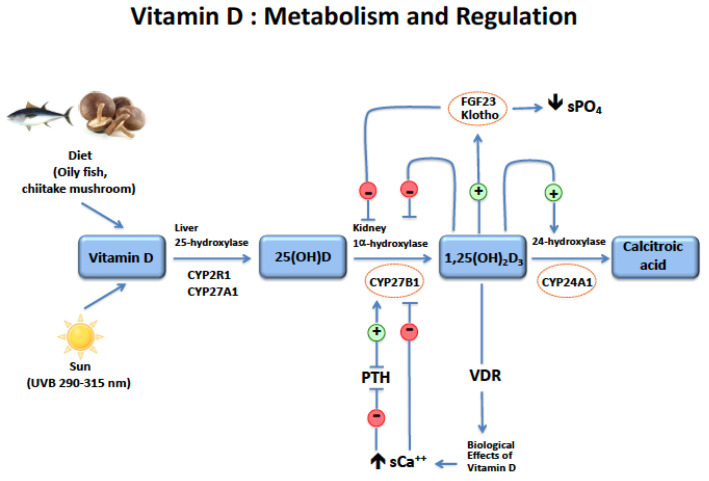
There are the following two sources of vitamin D in the diet: either D3, which can be found in oily fish, or D2, which is present in some kinds of mushrooms. Additionally, the other source is from the conversion of 7-dehydrocholesterol into vitamin D in the skin by the sun through a specific UVB light (290–315 nm wavelength). This is the major source, providing more than 70–80% of our daily vitamin D needs. These Vitamin Ds (D2 and D3) are transformed in 25(OH)D in the liver by the following two enzymes: CYP2R1 and CYP27A1. The two enzymes are down-regulated by uremic toxins and high PTH. Then, 25 vitamin D is converted in the kidney to 1,25-dihydroxyvitamin D3 [1,25-(OH)2D3] by 1α-hydroxylase or CYP27B1. Additionally, lastly, 1,25-dihydroxyvitamin D is degraded to calcitroic acid by the 24-hydroxylase, or CYP24A1. CYP27B1 is tightly regulated by PTH and calcium. PTH stimulates it and calcium inhibits it, but calcium also negatively regulates PTH. Calcitriol binds to the vitamin D receptor (VDR) and stimulates intestinal calcium absorption, thus controlling PTH and CYP27B1 activity. In addition, 1,25-(OH)2D3 stimulates FGF23, which with its interaction with klotho increases the urinary excretion of phosphate and has negative feedback on CYP27B1. Calcitriol also directly inhibits 1α-hydroxylase activity and stimulates 24-hydroxylase and its catabolism.

**Figure 2 metabolites-12-00266-f002:**
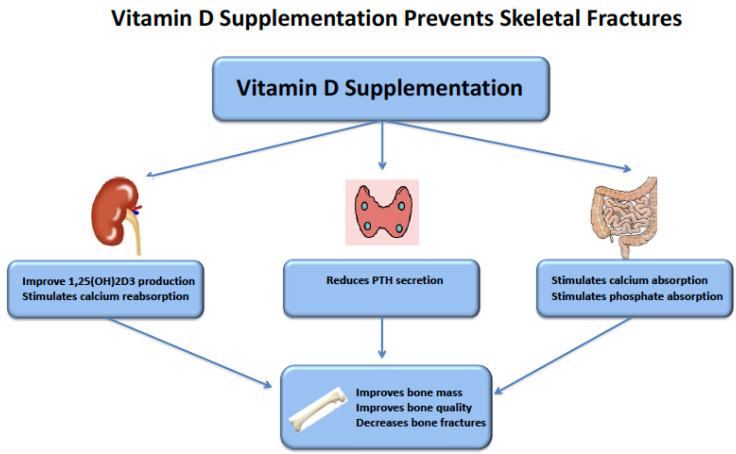
Plausible effects of vitamin D supplementation on skeletal fracture prevention. Natural vitamin D supplementation, either ergocalciferol or cholecalciferol, after their transformation in the liver into 25(OH)D2 or 25(OH)D3, favors the synthesis of active vitamin D (1,25(OH)2D3 and stimulates calcium reabsorption. It reduces PTH synthesis through the upregulation of parathyroid calcium sensing receptor and vitamin D receptor. It stimulates intestinal calcium absorption through the upregulation of TRPV6 (Transient Receptor Potential Cation Channel Subfamily Vanilloid Member 6) and intestinal phosphate absorption through the upregulation of sodium-phosphate cotransporter NPT2b. Altogether, these mechanisms result in an improved bone mass, quality, and strength and a reduced the risk of fractures.

**Table 1 metabolites-12-00266-t001:** Randomized clinical trials evaluating the effect of nutritional vitamin D supplementation on several mineral and bone metabolism parameters in patients with chronic kidney disease stage 3–5.

Number	Duration	Number of Patients	CKD Stage	Treatment	Main Results	Authors
**Bone Mineral Density**						
1	2 years	610	3 and 4	1200 mg of calcium + 800 IU of vitamin D3 versus placebo	Loss of BMD at the distal radius was reduced	Bosworth C. 2012 DECALYOS Study [69]
2	11 months	193	Kidney transplanted patients	4000 IU/day of vitamin D Versus placebo	Bone loss at the LS in treated subjects was attenuated (−0.2 versus −1.9%)	Tsujita M. 2021 [70]
**Circulating Biomarkers of Bone Metabolism**						
3	1 month	40	3 to 4	300,000 IU/month of vitamin D3 versus placebo	Decrease of PTH roughly from 368 to 279 pg/mL in the treated group	Dogan E. 2008 [71]
4	3 months	34	3 to 4	50,000 IU/week of vitamin D3 versus placebo	PTH showed a trend of reduction in the treated group	Chandra P. 2008 [72]
5	6 months	-	CKD diabetic patients	2000 IU/day versus 40,000 IU/day	Significant reduction in BSAP	Mager D.R. 2017 [73]
6	4 months	120	3 to 4	300,000 IU vitamin D3 every 2 months versus placebo	No significant change in serum sclerostin levels	Yadav A. 2018 [74]
7	4 months	120	3 to 4	300,000 IU vitamin D3 every 2 months versus placebo	BSAP decreased by 29 ng/mL CTX-1 decreased by 18 ng/mL No change in FGF23	Yadav A. 2017 [75]

Abbreviations: BMD, bone mineral density; LS, lumbar spine; PTH, parathyroid hormone; BSAP, bone-specific alkaline phosphatases; CTX-1, C-terminal telopeptide of type 1 collagen; FGF23, fibroblast growth factor 23; CKD, chronic kidney disease.

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
