# Peer review of "Bone Fragility in Chronic Kidney Disease Stage 3 to 5: The Use of Vitamin D Supplementation"

_metabolites, 2022, doi:10.3390/metabo12030266_

Round 1
Reviewer 1 Report
This is a nice paper. However, I have some comments.
The findings from this paper are excellent and worthy to review.
This manuscript contained some question s described below.
I think this paper is interesting, this revie w contributes to future's clinical
medicine largely.
I have some questions from a point of view of clinical medicine.
I apologize for the delay in sending me your comments.
What tests should be used as markers when evaluating bone fragility?
Is it a blood test, a bone density test, or an imaging test?
If so, what criteria should we administer VitD?
Author Response
Answers to Reviewer 1
First, we would like to thank the Reviewer for the kind introductory words and for the pertinent comments and suggestions.
Dear editors:
It is a great honor and pleasure for me to be invited as the reviewer for this important manuscript entitled “Bone Fragility in Chronic Renal Disease Stage 3 to 5: The use of Vitamin D Supplementation”. Pablo Antonio URENA TORRES and coauthors comprehensively reviewed therapeutic effects of Vitamin D supplementation for bone events in chronic kidney diseases (CKD) and clinical values. This study topic is interesting and novel, attributing to Prof. Martine COHEN SOLAL’s long-term efforts and contributions in this scientific field. Although the article is well-written, I have a number of comments concerning this study:
- The title of the article “Chronic Renal Disease” could be rephrase as “Chronic Kidney Diseases” to uniform the term of CKD throughout the manuscript.
We have now rephrased the title and replace “renal” by “kidney”
- Line 56: The word “CKD” should be spelled out.
The term CKD has been now spelled out as “chronic kidney disease”
- The term “25OHD” should be replaced by “25(OH)D” in Line 123 and Line 158 and so on.
We have now corrected and uniformized this term throughout all the text
- Line 168: “1,25OH2D3” should be rephrased as “1,25(OH)2D3”. Please check this throughout the manuscript.
We have now corrected and uniformized this term throughout all the text
Line 133: “24,25OH2D,” should be rephrased as “24,25(OH)2D3” . Please check this throughout the manuscript.
We have now corrected and uniformized this term throughout all the text
- Line 360: The section of conclusions should be rephrased. All the references should be removed. Authors should clarify the optimal dose/range of daily supplement and the kind of vitamin D3 (active or non-active). I agree with Line 342: “In the VITAL study, 2,000 IU vitamin D3 daily for 5.3 years did not induce any increase in the risk of kidney stone in apparently healthy subjects with a very satisfying mean baseline serum 25OHD concentration (31 ng/mL) [82].” In light of the nature of fat-soluble vitamins, the daily dose higher than 2,000 IU vitamin D3 may be harmful, especially for diverse population.
We have deleted all reference from the Conclusion section. The Conclusion section reads now : “Insufficient circulating levels of vitamin D is commonly observed in CKD patients, which favors SHPT and fragile bones. Although, international guidelines recommend supplementing CKD patients with nutritional vitamin D as in the general population. Generally, with 800-1400 IU/day of cholecalciferol or ergocalciferol and without surpassing 2,000 IU/day, and to achieve a serum 25(OH)D levels > 20 ng/ml. We believe that this level of circulating 25(OH)D might not be sufficient to prevent PTH raising and bone abnormalities in CKD as suggested by recent cohort studies where more than two third of CKD patients had controlled serum PTH levels when 25(OH)D levels were higher than 40 ng/ml. Similar findings have also been reported following the supplementation with ER calcifediol. Achieving such circulating 25(OH)D levels (50-75 ng/ml) seems to be safe, since they do not increase the risk of hypercalcemia, hyperphosphatemia, and urolithiasis. However, these promising results need further investigation looking at the impact of different natural vitamin D compounds on the risk of fractures, CKD progression and mortality”
- The viewpoint of the conclusion should be more conservative. To reduce the risk of vascular calcification, those CKD patients with hyperphosphatemia and hypercalcemia, calcitriol or another vitamin D sterol be reduced or stopped (1B), according to KDIGO guidelines. In patients with CKD G3a–G5D, they suggest that 25(OH)D (calcidiol) levels might be measured. In adult patients with CKD G3a–G5 not on dialysis, they suggest that calcitriol and vitamin D analogs not be
We have now rephrased the Conclusion section and be more cautious as the Reviewers suggested.
“Insufficient circulating levels of vitamin D is commonly observed in CKD patients, which favors SHPT and fragile bones. Although international guidelines recommend supplementing CKD patients with nutritional vitamin D as in the general population. Generally, with 800-1400 IU/day of cholecalciferol or ergocalciferol and without surpassing 2,000 IU/day, and to achieve a serum 25(OH)D levels > 20 ng/ml. We believe that this level of circulating 25(OH)D might not be sufficient to prevent PTH raising and bone abnormalities in CKD as suggested by recent cohort studies where more than two third of CKD patients had controlled serum PTH levels when 25(OH)D levels were higher than 40 ng/ml. Similar findings have also been reported following the supplementation with ER calcifediol. Achieving such circulating 25(OH)D levels (50-75 ng/ml) seems to be safe since they do not increase the risk of hypercalcemia, hyperphosphatemia, and urolithiasis, but in these studies the risk of cardiovascular calcifications was not evaluated. Additionally, as stated by KDIGO guidelines, to reduce the risk of vascular calcification, those CKD patients with hyperphosphatemia and hypercalcemia, calcitriol or another vitamin D sterol should be reduced or stopped. Therefore, the promising results observed with ER calcifediol need further investigation looking at the impact of different natural vitamin D compounds on the risk of fractures, CKD progression, cardiovascular calcifications, and mortality.
Reviewer 2 Report
Dear editors:
It is a great honor and pleasure for me to be invited as the reviewer for this important manuscript entitled “Bone Fragility in Chronic Renal Disease Stage 3 to 5: The use of Vitamin D Supplementation”. Pablo Antonio URENA TORRES and coauthors comprehensively reviewed therapeutic effects of Vitamin D supplementation for bone events in chronic kidney diseases (CKD) and clinical values. This study topic is interesting and novel, attributing to Prof. Martine COHEN SOLAL’s long-term efforts and contributions in this scientific field. Although the article is well-written, I have a number of comments concerning this study:
- The title of the article “Chronic Renal Disease” could be rephrase as “Chronic Kidney Diseases” to uniform the term of CKD throughout the manuscript.
- Line 56: The word “CKD” should be spelled out.
- The term “25OHD” should be replaced by “25(OH)D” in Line 123 and Line 158 and so on.
- Line 168: “1,25OH2D3” should be rephrased as “1,25(OH)2D3”. Please check this throughout the manuscript.
- Line 133: “24,25OH2D,” should be rephrased as “24,25(OH)2D3” . Please check this throughout the manuscript.
- Line 360: The section of Conclusions should be rephrased. All the references should be removed. Authors should clarify the optimal dose/range of daily supplement and the kind of vitamin D3 (active or non-active). I agree with Line 342: “In the VITAL study, 2,000 IU vitamin D3 daily for 5.3 years did not induce any increase in the risk of kidney stone in apparently healthy subjects with a very satisfying mean baseline serum 25OHD concentration (31 ng/mL) [82].” In light of the nature of fat-soluble vitamins, the daily dose higher than 2,000 IU vitamin D3 may be harmful, especially for diverse population.
- The viewpoint of the conclusion should be more conservative. To reduce the risk of vascular calcification, those CKD patients with hyperphosphatemia and hypercalcemia, calcitriol or another vitamin D sterol be reduced or stopped (1B), according to KDIGO guidelines. In patients with CKD G3a–G5D, they suggest that 25(OH)D (calcidiol) levels might be measured. In adult patients with CKD G3a–G5 not on dialysis, they suggest that calcitriol and vitamin D analogs not be
routinely used (2C).
Thank you for giving me the opportunity to review this interesting article. After minor revision, this important review article should be published as soon as possible.
Sincerely,
Author Response
Answers to Reviewer 1
First, we would like to thank the Reviewer for the kind introductory words and for the pertinent comments and suggestions.
Dear editors:
It is a great honor and pleasure for me to be invited as the reviewer for this important manuscript entitled “Bone Fragility in Chronic Renal Disease Stage 3 to 5: The use of Vitamin D Supplementation”. Pablo Antonio URENA TORRES and coauthors comprehensively reviewed therapeutic effects of Vitamin D supplementation for bone events in chronic kidney diseases (CKD) and clinical values. This study topic is interesting and novel, attributing to Prof. Martine COHEN SOLAL’s long-term efforts and contributions in this scientific field. Although the article is well-written, I have a number of comments concerning this study:
- The title of the article “Chronic Renal Disease” could be rephrase as “Chronic Kidney Diseases” to uniform the term of CKD throughout the manuscript.
We have now rephrased the title and replace “renal” by “kidney”
- Line 56: The word “CKD” should be spelled out.
The term CKD has been now spelled out as “chronic kidney disease”
- The term “25OHD” should be replaced by “25(OH)D” in Line 123 and Line 158 and so on.
We have now corrected and uniformized this term throughout all the text
- Line 168: “1,25OH2D3” should be rephrased as “1,25(OH)2D3”. Please check this throughout the manuscript.
We have now corrected and uniformized this term throughout all the text
Line 133: “24,25OH2D,” should be rephrased as “24,25(OH)2D3” . Please check this throughout the manuscript.
We have now corrected and uniformized this term throughout all the text
- Line 360: The section of Conclusions should be rephrased. All the references should be removed. Authors should clarify the optimal dose/range of daily supplement and the kind of vitamin D3 (active or non-active). I agree with Line 342: “In the VITAL study, 2,000 IU vitamin D3 daily for 5.3 years did not induce any increase in the risk of kidney stone in apparently healthy subjects with a very satisfying mean baseline serum 25OHD concentration (31 ng/mL) [82].” In light of the nature of fat-soluble vitamins, the daily dose higher than 2,000 IU vitamin D3 may be harmful, especially for diverse population.
We have deleted all reference from the Conclusion section. The Conclusion section reads now : “Insufficient circulating levels of vitamin D is commonly observed in CKD patients, which favors SHPT and fragile bones. Although, international guidelines recommend supplementing CKD patients with nutritional vitamin D as in the general population. Generally, with 800-1400 IU/day of cholecalciferol or ergocalciferol and without surpassing 2,000 IU/day, and to achieve a serum 25(OH)D levels > 20 ng/ml. We believe that this level of circulating 25(OH)D might not be sufficient to prevent PTH raising and bone abnormalities in CKD as suggested by recent cohort studies where more than two third of CKD patients had controlled serum PTH levels when 25(OH)D levels were higher than 40 ng/ml. Similar findings have also been reported following the supplementation with ER calcifediol. Achieving such circulating 25(OH)D levels (50-75 ng/ml) seems to be safe, since they do not increase the risk of hypercalcemia, hyperphosphatemia, and urolithiasis. However, these promising results need further investigation looking at the impact of different natural vitamin D compounds on the risk of fractures, CKD progression and mortality”
- The viewpoint of the conclusion should be more conservative. To reduce the risk of vascular calcification, those CKD patients with hyperphosphatemia and hypercalcemia, calcitriol or another vitamin D sterol be reduced or stopped (1B), according to KDIGO guidelines. In patients with CKD G3a–G5D, they suggest that 25(OH)D (calcidiol) levels might be measured. In adult patients with CKD G3a–G5 not on dialysis, they suggest that calcitriol and vitamin D analogs not be
We have now rephrased the Conclusion section and be more cautious as the Reviewers suggested.
“Insufficient circulating levels of vitamin D is commonly observed in CKD patients, which favors SHPT and fragile bones. Although international guidelines recommend supplementing CKD patients with nutritional vitamin D as in the general population. Generally, with 800-1400 IU/day of cholecalciferol or ergocalciferol and without surpassing 2,000 IU/day, and to achieve a serum 25(OH)D levels > 20 ng/ml. We believe that this level of circulating 25(OH)D might not be sufficient to prevent PTH raising and bone abnormalities in CKD as suggested by recent cohort studies where more than two third of CKD patients had controlled serum PTH levels when 25(OH)D levels were higher than 40 ng/ml. Similar findings have also been reported following the supplementation with ER calcifediol. Achieving such circulating 25(OH)D levels (50-75 ng/ml) seems to be safe since they do not increase the risk of hypercalcemia, hyperphosphatemia, and urolithiasis, but in these studies the risk of cardiovascular calcifications was not evaluated. Additionally, as stated by KDIGO guidelines, to reduce the risk of vascular calcification, those CKD patients with hyperphosphatemia and hypercalcemia, calcitriol or another vitamin D sterol should be reduced or stopped. Therefore, the promising results observed with ER calcifediol need further investigation looking at the impact of different natural vitamin D compounds on the risk of fractures, CKD progression, cardiovascular calcifications, and mortality.

Reviewer 3 Report
In the manuscript titled: “Bone Fragility in Chronic Renal Disease Stage 3 to 5: The use of 2 Vitamin D Supplementation”, the authors revised the contribution about bone fragility in CKD by vitamin D lacking.
- Although the manuscript is well written, some grammatical and spelling mistakes should be addressed, please, check this.
- In the abstract, “Its pathophysiology includes the reduction of active vitamin D metabolites, phosphate accumulation, decreased intestinal calcium absorption, decreased renal alpha klotho production, and elevated fibroblast growth factor 23 (FGF23) levels, all of them contributing firstly to secondary hyperparathyroidism and ultimately to micro and macrostructural bone changes, which lead to low bone mineral density and increased risk of fracture” Please add a point after (FGF23) levels because is very long. In these lines, the authors should eliminate the second “decreased”.
- In the lines 200-203: “However, this meta-analysis included studies that recruited an inadequate 200 number of participants or used either very low daily doses or very high intermittent 201 (annually) doses, two actions known to have either no effect (very low doses) or even 202 detrimental effects (very high intermittent doses) on the risk of fracture.” This is not clear. What are the molecular effects of consuming high or low levels of vitamin D?
- In Table 1, the first column should be the last, please modify.
- Authors should add a discussion section because a lot of numerical data is present and often not adequately discussed.
- The authors should add a figure, proposing how vitamin D consumption is beneficial to improve fractures in CKD.
Author Response
Answers to Reviewer 2
First, we would like to thank the Reviewer for the kind introductory words and for the pertinent comments and suggestions.
Reviewer 2
In the manuscript titled: “Bone Fragility in Chronic Renal Disease Stage 3 to 5: The use of 2 Vitamin D Supplementation”, the authors revised the contribution about bone fragility in CKD by vitamin D lacking.
- Although the manuscript is well written, some grammatical and spelling mistakes should be addressed, please, check this.
We thank the Reviewer for this comment. We have read again the manuscript a correct some of the grammar mistake we made. However, we hope it will be read by an English Expert from the Editorial office.
- In the abstract, “Its pathophysiology includes the reduction of active vitamin D metabolites, phosphate accumulation, decreased intestinal calcium absorption, decreased renal alpha klotho production, and elevated fibroblast growth factor 23 (FGF23) levels, all of them contributing firstly to secondary hyperparathyroidism and ultimately to micro and macrostructural bone changes, which lead to low bone mineral density and increased risk of fracture” Please add a point after (FGF23) levels because is very long. In these lines, the authors should eliminate the second “decreased”.
We have corrected the Abstract section as the Reviewer suggested. “Its pathophysiology includes the reduction of active vitamin D metabolites, phosphate accumulation, decreased intestinal calcium absorption, renal alpha klotho production, and elevated fibroblast growth factor 23 (FGF23) levels. Altogether, these factors contribute firstly to secondary hyperparathyroidism and ultimately to micro- and macrostructural bone changes, which lead to low bone mineral density and increased risk of fracture”.
- In the lines 200-203: “However, this meta-analysis included studies that recruited an inadequate 200 number of participants or used either very low daily doses or very high intermittent 201 (annually) doses, two actions known to have either no effect (very low doses) or even 202 detrimental effects (very high intermittent doses) on the risk of fracture.” This is not clear. What are the molecular effects of consuming high or low levels of vitamin D?
We have now tried to clarify this phrase. Indeed, the molecular effects of consuming high or low vitamin D are well known. Vitamin D modulates the transcription of more than 1,000 genes and thus the expressed molecules, which was not the scope of this article. What is not clear is the mechanism by which excessive vitamin D consumption is associated with increased risk of falls and fractures. The phrase reads now:
“However, this meta-analysis included studies that recruited a relatively low number of participants or used either very low daily doses (400 UI/day) or very high intermittent (300,000-500,000 IU/year) doses, two actions known to have either no effect (very low doses) or even detrimental effects (very high intermittent doses) on the risk of falls and fractures through the mechanisms not yet understood [66]”.
- In Table 1, the first column should be the last, please modify.
This point has now been modified.
- Authors should add a discussion section because a lot of numerical data is present and often not adequately discussed.
We disagree with the Reviewer comment. This is already a review article, a large discussion of numerous studies. We do not think that it is appropriate adding another section.
- The authors should add a figure, proposing how vitamin D consumption is beneficial to improve fractures in CKD.
We agree with this comment and have added a schematic representation of the plausible effects of vitamin D supplementation on skeletal fracture prevention.

Round 2
Reviewer 3 Report
Although most of the changes were made, the authors did not conclude the critical sections of the article, which is the main weakness of the manuscript.
Author Response
Reviewer 3
To the author
This is a nice paper. However, I have some comments. The findings from this paper are excellent and worthy to review. This manuscript contained some questions described below. I think this paper is interesting, this review contributes to future's clinical medicine largely.
I have some questions from a point of view of clinical medicine.
I apologize for the delay in sending me your comments.
What tests should be used as markers when evaluating bone fragility?
Is it a blood test, a bone density test, or an imaging test?
As recommended by the recent revised 2021 KDIGO guidelines, when evaluating CKD-MBD and bone fragility, for the blood tests, serum calcium, phosphate and PTH should be assessed at the beginning and at least every three months in CKD patients. Circulating 25(OH)D levels should be assessed at the dialysis initiation and then once a year or more frequent in case of vitamin D supplementation or in case of evaluating an increased serum PTH level.
A Bone mineral density is recommended for the evaluation of bone fragility in CKD only if the results will impact therapeutical decision.
No imaging test, such as standard X-Ray, TDM, MRI, etc, is clearly recommended, and is left to the physician discretion.
If so, what criteria should we administer VitD?
The only criterium for the vitamin D supplementation according to KDIGO guidelines is when circulating 25(OH)D level is inferior to 20 ng/ml.
Answers to Second Round for Reviewer 3
Although most of the changes were made, the authors did not conclude the critical sections of the article, which is the main weakness of the manuscript.
Thanks for this comment and we have now added a short conclusion, we hope in the critical section of the article as considered by the Reviewer.
Round 3
Reviewer 3 Report
The manuscript was improved